# Comparative Analysis of Flame Propagation and Flammability Limits of CH₄/H₂/Air Mixture with or without Nanosecond Plasma Discharges

Ghazanfar Mehdi, Maria Grazia De Giorgi *, Sara Bonuso, Zubair Ali Shah, Giacomo Cinieri and Antonio Ficarella

Department of Engineering for Innovation, University of Salento, Via per Monteroni, 73100 Lecce, Italy
* Correspondence: mariagrazia.degiorgi@unisalento.it

**Abstract:** This study investigates the kinetic modeling of $CH_4/H_2/Air$ mixture with nanosecond pulse discharge (NSPD) by varying $H_2/CH_4$ ratios from 0 to 20% at ambient pressure and temperature. A validated version of the plasma and chemical kinetic mechanisms was used. Two numerical tools, ZDPlasKin and CHEMKIN, were combined to analyze the thermal and kinetic effects of NSPD on flame speed enhancement. The addition of $H_2$ and plasma excitation increased flame speed. The highest improvement (35%) was seen with 20% $H_2$ and 1.2 mJ plasma energy input at $\phi = 1$. Without plasma discharge, a 20% $H_2$ blend only improved flame speed by 14% compared to 100% CH4. The study found that lean conditions at low flame temperature resulted in significant improvement in flame speed. With 20% $H_2$ and NSPD, flame speed reached 37 cm/s at flame temperature of 2040 K at $\phi = 0.8$. Similar results were observed with 0% and 5% $H_2$ and a flame temperature of 2200 K at $\phi = 1$. Lowering the flame temperature reduced $NO_x$ emissions. Combining 20% $H_2$ and NSPD also increased the flammability limit to $\phi = 0.35$ at a flame temperature of 1350 K, allowing for self-sustained combustion even at low temperatures.

**Keywords:** flame propagation; nanosecond plasma discharges; lean burning; ZDPlaskin; CHEMKIN

## 1. Introduction

Combustion is a crucial factor in air transportation due to the high energy density of liquid fuels. However, the low efficiency of current aeroengines and the production of harmful emissions contributing to climate change are pressing issues. To comply with strict emission regulations set by CAEP (Committee on Aviation Environmental Protection) and improve fuel efficiency, various international organizations are exploring the concept of lean combustors.

Lean fuel burning is an effective solution for reducing $NO_x$ emissions by lowering flame temperature. However, these low temperature flames are prone to critical instabilities that can lead to re-ignition and flame blowout issues [1,2]. To address issues with methane combustion, the addition of a more reactive and cleaner fuel such as hydrogen could be a practical solution [3]. Blending methane with hydrogen has been shown to enhance performance and reduce emissions without modifying existing combustors [4]. Hydrogen is a carbon-free fuel with low ignition energy, a wide flammability range, fast flame propagation, and high reactivity [3]. Several studies in the past [3–6] have focused on the impact of hydrogen on the flame speed of $CH_4/H_2$ mixtures. Halter et al. [5] studied the effect of hydrogen content and inlet pressure on the laminar flame speed of $CH_4/H_2$ flames, with results indicating that the laminar flame speed improved with increasing hydrogen content and decreased with increasing inlet pressure.

Mandilas et al. [6] studied the impact of hydrogen on iso-octane-air and methane mixtures in both laminar and turbulent conditions. They found that using hydrogen led to earlier flame instabilities but improved laminar flame speed at lean limits in turbulent

combustion. Adding hydrogen to methane slightly improved reactivity at lean conditions, but also increased complexities, safety issues, and thermoacoustic instabilities [3–6]. Flame speed was slightly better at lean compared to rich conditions [4]. Non-thermal plasma combustion can improve flame stability, flame speed, and lean blowout limits. NTP enhances combustion through kinetic, thermal and momentum effects [7]. NTP improves combustion through three mechanisms: kinetic (creation of active particles from fuel decomposition), thermal (increased fuel/air mixture temperature), and momentum (ionic wind and flow motion from electro-hydrodynamic forces) [8].

Among NTP technologies, nanosecond plasma discharge (NSPD) has gained attention due to its ability to effectively produce excited states and active particles [9,10]. NSPD also rapidly heats the gas, which accelerates combustion [11,12]. Despite numerous research studies on NTP combustion [1], commercialization is only possible with the development of accurate numerical models for plasma chemistry in combustion. Our group [12–15] has studied CH4/air mixtures with NSPD for flame propagation and ignition enhancement. We compared ignition delay, flame speed, and flammability limits under different conditions and found that NSPD improved ignition, flame propagation, and flammability limits due to the production of neutral radicals and increased mixture temperature. The improvements were primarily due to NSPD's kinetic effects. Prior studies on NSPD have individually considered H2/air and CH4/air mixtures.

While initial studies have explored the kinetics of NSPD, a comprehensive understanding of plasma mechanisms for $CH_4/H_2/air$ mixtures is still lacking. It has been shown that the evolution of active particles over time provides the most accurate analysis of plasma kinetics [16,17].

This paper presents a study of $CH_4/H_2/air$ with nanosecond plasma discharge. There is currently no numerical study available on methane blended hydrogen plasma-assisted combustion. Both plasma and combustion kinetics were analyzed using validated mechanisms and compared to previously published experimental data. The impact of NSPD and hydrogen content on flame propagation and flammability limits in methane/air mixtures was studied. A comparative analysis of flame speed enhancement with and without plasma actuation was performed using different methane blended hydrogen ratios.

## 2. Numerical Procedure and Kinetic Modelling

### 2.1. Numerical Procedure

Numerical analyses were conducted using two solvers: ZDPlasKin (0D Plasma kinetic solver) [18] and CHEMKIN (Chemical kinetic solver) [19]. The methodology is shown in Figure 1 and explained in [13]. ZDPlasKin was used to analyze the kinetic and thermal effects of NSPD in $CH_4/H_2/Air$ mixture. BOLSIG+ was linked to ZDPlasKin to predict the temporal evolution of excitation states and the reactions producing free radicals/active particles. It has been assumed that the non-equilibrium plasma created from a $CH_4/H_2/air$ mixture at atmospheric pressure is uniformly distributed, which is a similar assumption to what was previously executed in [13]. Although the nanosecond pulsed plasma combustion process is three-dimensional and not homogeneous, we used a simplified homogeneous model. To investigate the effects of plasma $CH_4/H_2/air$ products on flame speed and flammability limits, we used the plasma products of $CH_4/H_2/air$ as the inlet domain of the reactor.

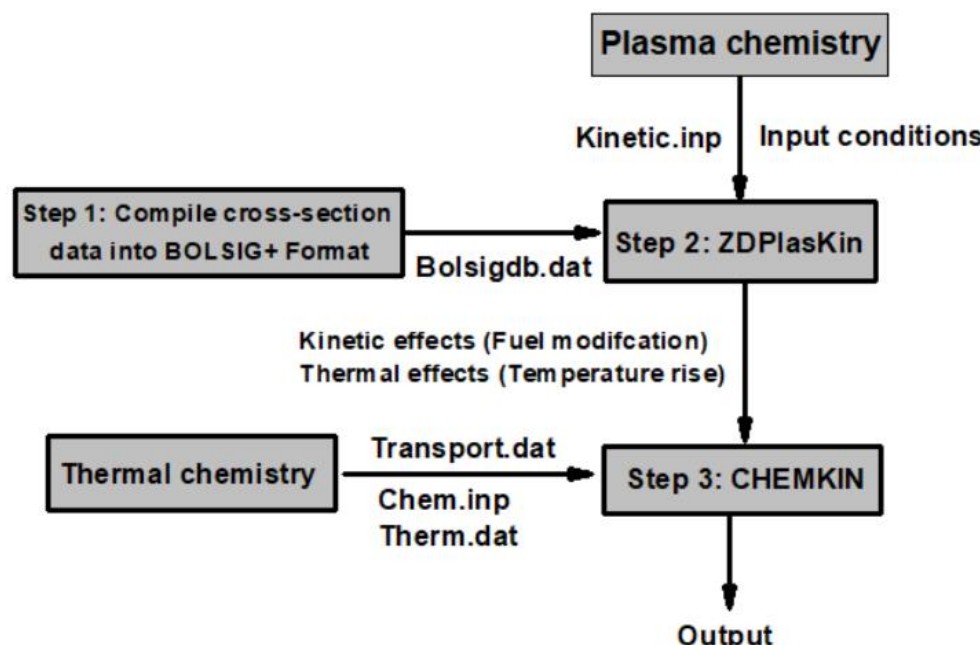

**Figure 1.** Flowchart for numerical analysis.

ZDPlasKin boundary conditions were set as ambient temperature and pressure, fixed EN and electron number density, and initial $CH_4/H_2/Air$ composition. The simplified homogeneous model was used as in [20]. ZDPlasKin simulation was performed using the integral mean value of $E_N$ obtained from experiments, about 200 Td over $10^{-6}$ s, as shown in Figure 2. Experimental setup and $E_N$ estimation are described in [13]. The gas temperature was predicted using equations from [18]. The adiabatic gas temperature was calculated from the energy conservation equation and reallocation of electrical power $P_{ext}$ to electron translational degree $P_{elec}$, gas internal degree $P_{chem}$, and gas $P_{gas}$:

$$Pext = Pgas + Pelec + Pchem \tag{1}$$

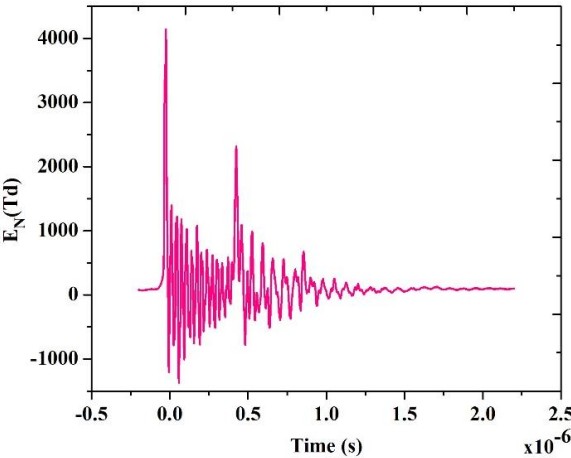

**Figure 2.** Experimental $E_N$ value used for numerical analysis [13].

The above equation can be described below.

$$Pext = e[Ne]veE \tag{2}$$

$$Pgas = \frac{1}{\gamma - 1} + \frac{d(NTgas)}{dt} \tag{3}$$

$$Pelec = \frac{3}{2} + \frac{d([Ne]Te)}{dt} \tag{4}$$

$$Pchem = \sum_{i}^{n} Qi + \frac{d[Ni]}{dt} \tag{5}$$

where $E$ is the reduced field, $e$ is the elementary charge, $T_e$ is the electron temperature, *ve* is the drift velocity of electrons $[N_e]$ is the electron density, $N$ is the total gas density, $\gamma = 1.2$ is the specific gas heat ratio and $Q_i$ is the potential energy of species $i$.

The results gained from the ZDPlaskin solver in terms of neutral and excited species (at a time of 0.5 ms because the residence time is too short that autoignition chemistry does not significantly influence the reactants compositions), and the gas temperature of the activated region were introduced into the CHEMKIN solver to investigate the combustion process.

A 1-D premixed laminar flame speed reactor was employed to analyze combustion characteristics, considering thermal diffusion and multicomponent diffusion options. The adaptive mesh parameters were set as CURV = 0.5 and GRAD = 0.05, with absolute and relative error criteria of $A_{TOL} = 1 \times 10^{-9}$ and $R_{TOL} = 1 \times 10^{-5}$, respectively. The total number of grid points used was typically 350–400. In this study, we have established that the calculation domain of the CHEMKIN reactor ranges from −2.0 cm upstream to 4.0 cm downstream with respect to the reactor and is sufficient to attain adiabatic equilibrium. Numerical analyses were performed at various fueling conditions based on $H_2/CH_4$ ratio ($x_{H2}$) with or without plasma actuation. Table 1 shows the mole fraction of $CH_4$, and $H_2$ reactants at equivalence ratio of 1.

**Table 1.** Reactants mole fraction of $CH_4/H_2/Air$ flames at plasma on and off conditions.

| Case No. | $H_2$ (%) | $CH_4$ | $H_2$ | $O_2$ | $N_2$ | Plasma (ON/OFF) |
|----------|-----------|--------|--------|--------|--------|-----------------|
| 1 | 0 | 0.0950 | / | 0.1900 | 0.7149 | OFF |
| 2 | 5 | 0.0935 | 0.0049 | 0.1894 | 0.7122 | OFF |
| 3 | 10 | 0.0917 | 0.0101 | 0.1885 | 0.7094 | OFF |
| 4 | 20 | 0.0879 | 0.0219 | 0.1869 | 0.7031 | OFF |
| 5 | 0 | 0.0950 | / | 0.1900 | 0.7149 | ON |
| 6 | 5 | 0.0935 | 0.0049 | 0.1894 | 0.7122 | ON |
| 7 | 10 | 0.0917 | 0.0101 | 0.1885 | 0.7094 | ON |
| 8 | 20 | 0.0879 | 0.0219 | 0.1869 | 0.7031 | ON |

*2.2. Plasma Kinetic Model*

A comprehensive literature review was conducted to develop an extended plasma kinetic mechanism for $CH_4/H_2/Air$ mixture. It consists of 161 species and 1382 plasma and gas-phase reactions, and includes ionization reactions, charged transfer reactions, dissociation reactions, excited species reactions, recombination reactions, relaxation reactions, and three-body recombination reactions. The mechanism also included 38 exciting species and 35 charged species. The relevant reactions were taken from [21–24]. The collision cross-sectional data were taken from the LXCat data source [13]. Further information can be found in the previously published study [13].

*2.3. Combustion Kinetic Model*

The NSPD generated kinetic effects (neutral radicals, active particles, excited species) and thermal effects were used to study the effect on flame speed in a CHEMKIN combustion model. The $CH_4/H_2/Air$ combustion kinetic model was created with an expanded version of the combustion mechanism, incorporating thermodynamics and transport data. The mechanism was updated from GRI-Mech v3.08 with ozone reactions [25] and updated hydrogen combustion mechanism including the excited species O(1D), OH(2+), $O_2(a^1g)$ [26]. A sub-model of the excited species OH* and CH* has also been added [27]. Furthermore, the reaction mechanism of ions and excited species of $CH_4/Air$ mixture was also considered [28].

### 3. Validation of Kinetic Models

The plasma kinetic model was validated using an experimental study in [29] by comparing the mole fraction of the decay process of O atoms. The plasma kinetic model was validated using an experimental study in [13]. The model accurately predicts the O atom mole fraction, in good agreement with experimental data.

The combustion kinetic model was validated using experiments by Coppens [30], Hermanns [31], the Konnov mechanism [32], and the San Diego mechanism [33]. The combustion kinetic mechanism was tested with $H_2$ and $CH_4$ fractions equal $x_{H2} = 0.05$ and $x_{CH4} = 0.95CH_4$ at different equivalence ratios. Figure 3 shows the validation results for burning velocities of $CH_4/H_2/Air$ mixture with 5%, 30%, and 40%. $H_2$ content. The model shows good agreement with experimental data compared to other mechanisms, with slightly lower values at rich burning conditions as seen in [30,31].

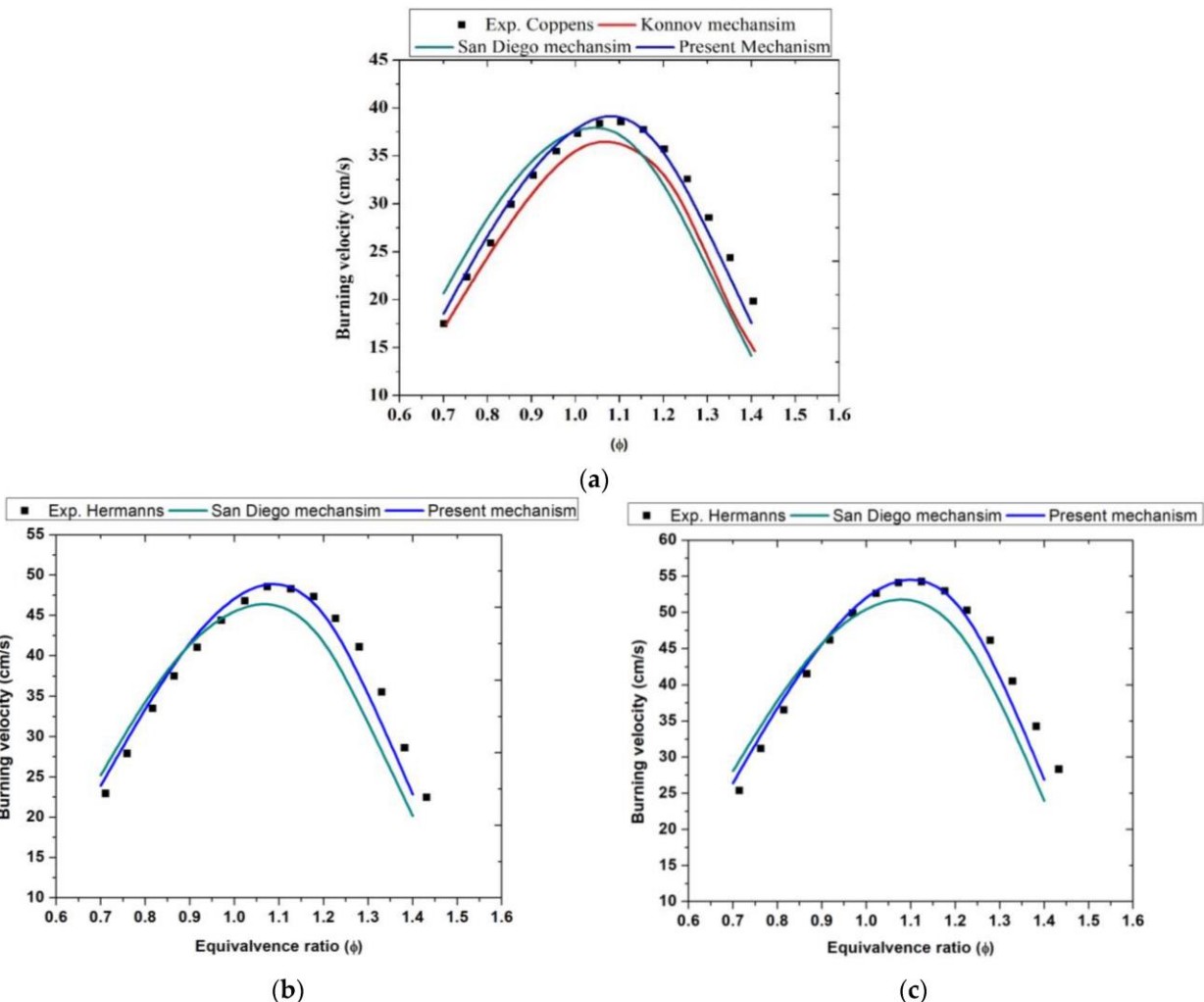

**Figure 3.** Validation: predicted values of burning velocities of $CH_4/H_2/Air$ mixture with a $H_2$ content of (**a**) 5%, (**b**) 30% and (**c**) 40%.

### 4. Results and Discussions

The present analysis was conducted under fixed plasma conditions: $E_N = 200$ Td, repetition frequency equal to 1000 Hz, and electron number density equal to $10^7$ cm$^{-3}$. The effect of varying $H_2$ contents ($x_{H2}$ from 0 to 0.2) on active particle production was studied in methane/air. Figure 4 shows the temporal evolution of active particles (H, OH, CH, and $CH_3$) as predicted by ZDPlaskin simulations under fixed plasma actuation conditions, only changing the $H_2$ content in the methane/air mixture. An increase in $H_2$ concentration

led to a significant improvement in the mole fraction of active species. The improvement in active particles production was linearly proportional to the rise in $H_2$ content, with the maximum concentration observed at 20% $H_2$. The rapid decomposition of $H_2^+$ into H, (E + $H_2^+$ => H + H) due to its simple molecular structure and high reactivity and the subsequent reactions with other intermediate species, led to increased concentration of active particles. The maximum mole fraction of H was 0.00704 (Figure 4a), which was almost twice the OH species equal to 0.0038 (Figure 4b). The maximum mole fraction of $CH_3$ was 0.006 (Figure 4c), slightly less than H but two orders higher than the molar fraction of CH (0.000041, Figure 4d).

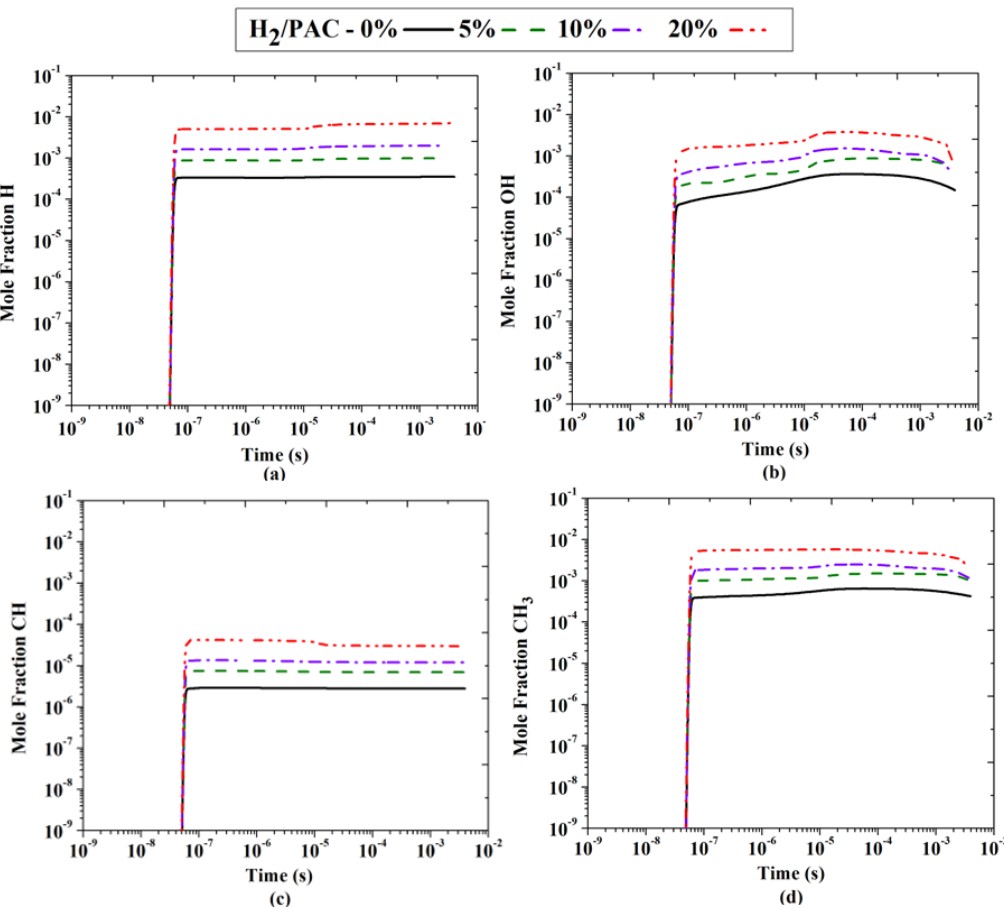

**Figure 4.** Temporal evolution of (**a**) H, (**b**) OH, (**c**) CH, (**d**) $CH_3$ concentrations at different contents of $H_2$ in methane/air mixture using fixed plasma actuation conditions.

The H atoms were produced during the decomposition process when electrons reacted with the ions of $CH^+$, $CH_2^+$, $CH_3^+$, $CH_4^+$, and $H_3O^+$. The primary reactions contributing to the production of H, CH, and $CH_3$ were E + $CH_4^+$ = $CH_3$ + H and E + $CH_3^+$ = CH + H + H (reaction rates: $10^{20}$ and $10^{21}$ $cm^3$ $s^{-1}$, respectively). The OH radicals were produced through the reaction O(1D) + $CH_4$ = $CH_3$ + OH (reaction rate: $10^{24}$ $cm^3$ $s^{-1}$).

As shown in Figure 5, the mole fraction of active species was significantly improved with an increase in $H_2$ content in the methane/air mixture. The highest mole fraction of O atoms was observed at a 20% $H_2$ content ($x_{H2}$ = 0.2) with a value of 0.0158 (Figure 5b). This was due to the decomposition of excited $O_2$ species when reacting with H atoms and $H_2O$ molecules, which increased in concentration due to the presence of H2 molecules in the methane/air mixture. The dominant reaction path was H + $O_2(V_4)$ => O + OH (reaction rate: 1023 $cm^3$/s). The O atoms produced began to reduce after $10^{-4}$ s, likely due to the short reactive time of O atoms leading to their consumption during recombination and

intermediate reactions. Similarly, ozone concentration improved as shown in Figure 5c, with a mole fraction of 0.00729, close to that of H atoms (0.00704) at a 20% $H_2$ content.

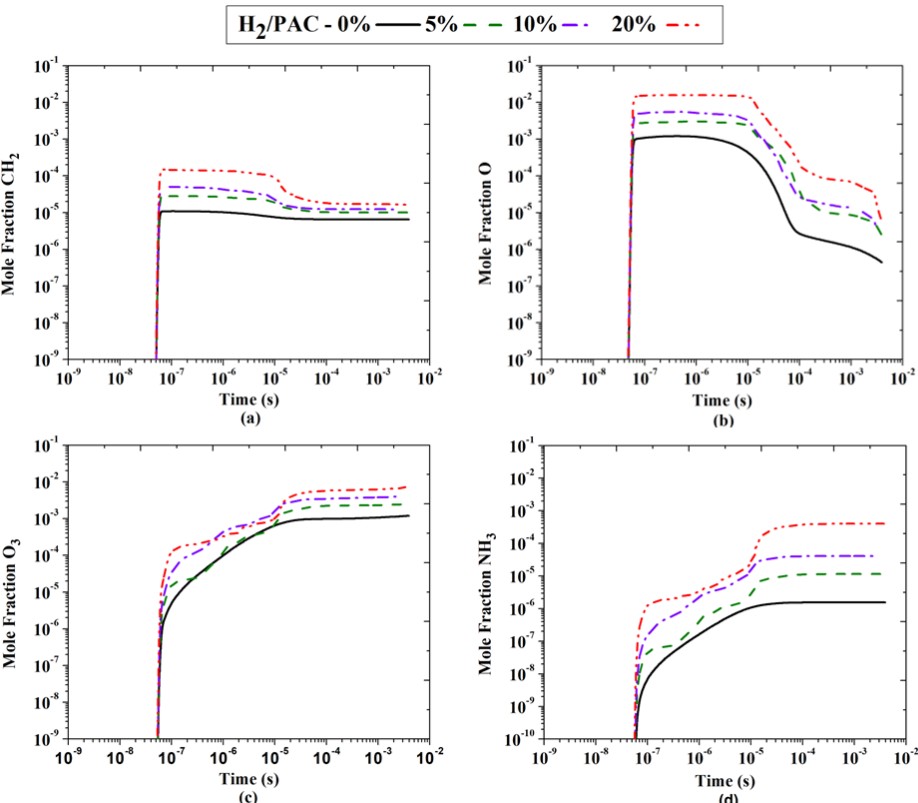

**Figure 5.** Temporal evolution of (**a**) $CH_2$, (**b**) O, (**c**) $O_3$, (**d**) $NH_3$ concentrations at different contents of $H_2$ in methane/air mixture using fixed plasma actuation conditions.

Ozone is primarily generated through the reaction of O atoms with molecular oxygen or with excited species of oxygen. The most significant reaction is $O + O_2 + N_2 = O_3 + N_2$, as it has a reaction rate of about 1023 $cm^3/s$. Nitrogen acts as a third body, removing excess energy. Ozone can also improve combustion analysis as it increases flame speed [25]. The temporal evolution of ammonia was also found to improve when nitrogen seed particle was added to methane-blended hydrogen.

The kinetic and thermal effects predicted by ZDPlasKin were introduced into Chemkin to investigate the flame speed and maximum flame temperature. The reactant mole fraction of active particles and excited species predicted by ZDPlasKin were added to Chemkin to account for kinetic effects. This was executed with a 0.5 ms residence time, which is too short to affect the autoignition chemistry and the reactant composition. The flame speed and peak flame temperature were investigated using a pre-mixed laminar flame speed reactor at different methane-blended hydrogen mixture compositions with or without NSPD.

Figure 6a showed that flame speed improved with increasing hydrogen content and plasma excitation. At stoichiometric mixture, adding hydrogen ($x_{H2} = 0.2$) to the methane/air mixture resulted in a 14% increase in flame speed ($\Delta s_L$). A further improvement of 35% was achieved with plasma discharge. At leaner condition ($\phi = 0.6$) and same $H_2$ fraction, $\Delta s_L$ was 16.7% without and 52% with plasma actuation. However, the same flame speed was observed for both cases of $x_{H2} = 0.2$ and $x_{H2} = 0.05$ with PAC at lean and stoichiometric conditions, similarly in case of $x_{H2} = 0.1$ and $x_{H2} = 0$ with PAC. It means the same range of flame speed could be reached by varying both hydrogen fraction and plasma discharge.

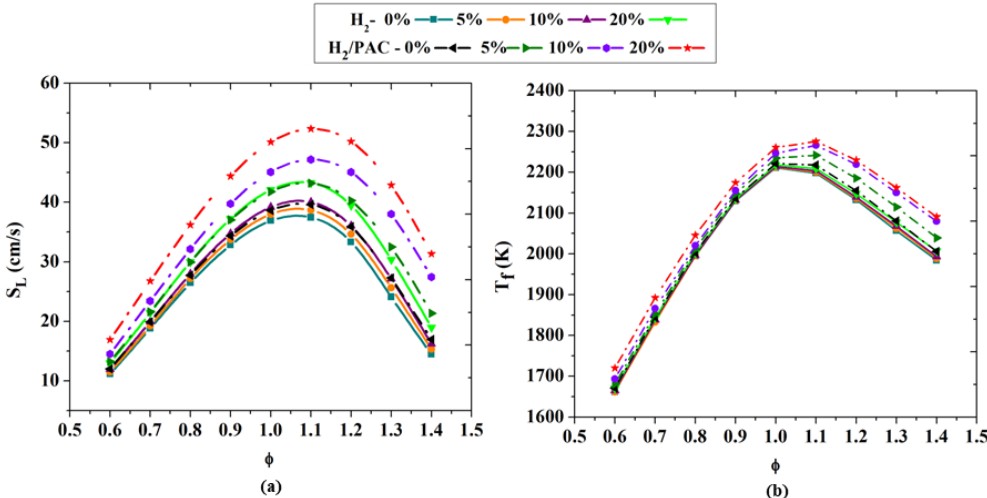

**Figure 6.** Comparison of (**a**) flame speed and (**b**) flame temperature at various equivalence ratios using different $H_2$ contents with or without NSPD.

Figure 6b shows the predicted peak flame temperature $T_f$ with or without plasma for various $H_2$ fractions. The results showed that increasing $H_2$ did not affect $T_f$ without plasma discharges. However, with plasma, $T_f$ was affected by $H_2$ at fixed operating conditions, especially for the rich mixture ($\Phi > 1$). A slightly increase in $T_f$ was found at lean and stoichiometric conditions.

The study found that lean conditions at low flame temperature resulted in significant improvement in flame speed. With 20% $H_2$ and NSPD, flame speed reached 37 cm/s at flame temperature of 2040 K at $\phi = 0.8$. Similar results were observed with 0% and 5% $H_2$ and a flame temperature of 2200 K at $\phi = 1$. It was observed that the same flame speed can be achieved at lean conditions by reducing $T_f$, leading to reduced $NO_x$ emissions.

Figure 7 compares the improvement in flame speed (%) at lean, stochiometric, and rich conditions with $x_{H2} = 0.2$ with or without NSPD. It was observed that the improvement trend was $\phi = 1.4 > \phi = 0.6 > \phi = 1$. At lean conditions ($\phi = 0.6$), adding $x_{H2} = 0.2$ improved flame speed by 15%, however, using both $x_{H2} = 0.2$ and plasma resulted in a more than 50% improvement. At rich conditions, the largest improvement was seen with plasma due to the increased fuel causing more active particles to be produced during NSPD".

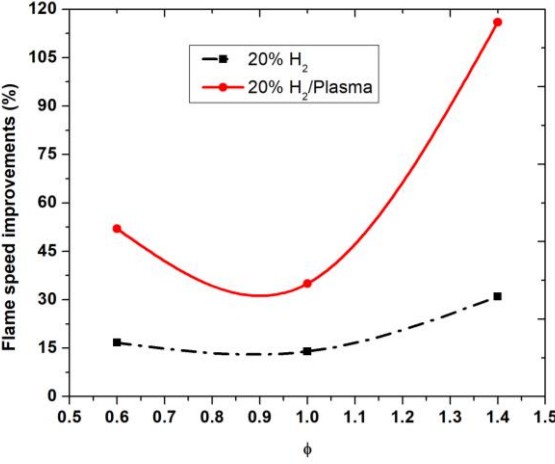

**Figure 7.** Comparative behavior of flame speed improvements (%) at lean, stoichiometric, and rich conditions.

Literature [26] showed that the molecular excited species oxygen $O_2(1\Delta)$ increased burning velocity by 1% without plasma. The reaction path $H_2 + O_2(1\Delta) = H + HO_2$ was

found to play a significant role. More than 5% of $O_2$ was converted to $O_2(1\Delta)$ in the presence of electric discharge at ambient pressure [34]. Thus, plasma discharge could produce significant amounts of $O_2(1\Delta)$. Figure 8 analyzed the role of $O_2(1\Delta)$ using hydrogen blends with or without plasma discharge. The results showed no change in $O_2(1\Delta)$ production with hydrogen blends alone at various equivalence ratios. However, with the use of plasma discharge, there was a significant rise in excited species production, especially at higher hydrogen content.

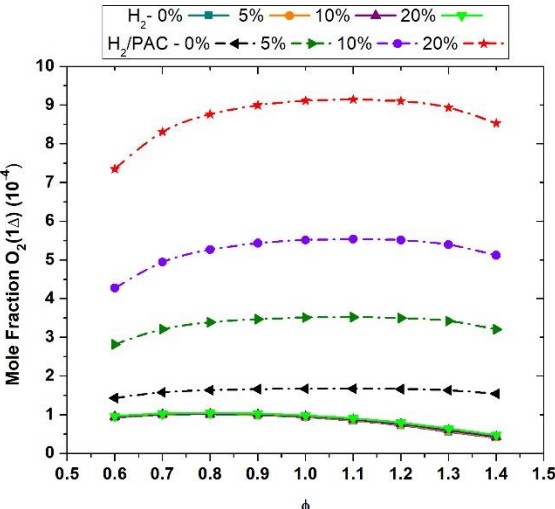

**Figure 8.** Comparison of molecular excited species $O_2(1\Delta)$ at various equivalence ratios using different $H_2$ contents with or without NSPD.

Impact of atomic excited species O(1D) on flame speed was studied using plasma and hydrogen blends (Figure 9). Results showed low O(1D) concentration increased with hydrogen and plasma, but still had minimal effect on combustion. However, it could be increased with the increase in plasma amplitude.

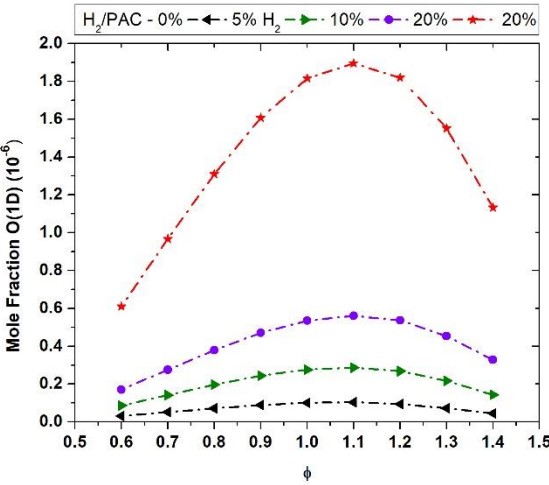

**Figure 9.** Comparison of atomic excited species O(1D) at various equivalence ratios using different $H_2$ contents with NSPD.

Free radicals such as O, H, and OH are active due to unpaired electrons and short lived in combustion [35]. They initiate chain reactions and branching. Figure 10a–d show mole fraction profiles of O, H, OH, and $CH_3$ using hydrogen blends with/without plasma discharge. Adding hydrogen increased O, H, and OH mole fractions, but decreased $CH_3$

slightly (Figure 10d). Using NSPD in hydrogen blends raised O, H, and OH concentrations and moved the reaction region upstream. CH$_3$ mole fraction was also slightly increased with plasma discharge. OH particles had the highest concentration at 0.009 mole fraction. Main reactions producing O, H, and OH particles are described as follows in Equations (6) and (7).

$$OH + H_2 = H + H_2O \tag{6}$$

$$H + O_2 = O + OH \tag{7}$$

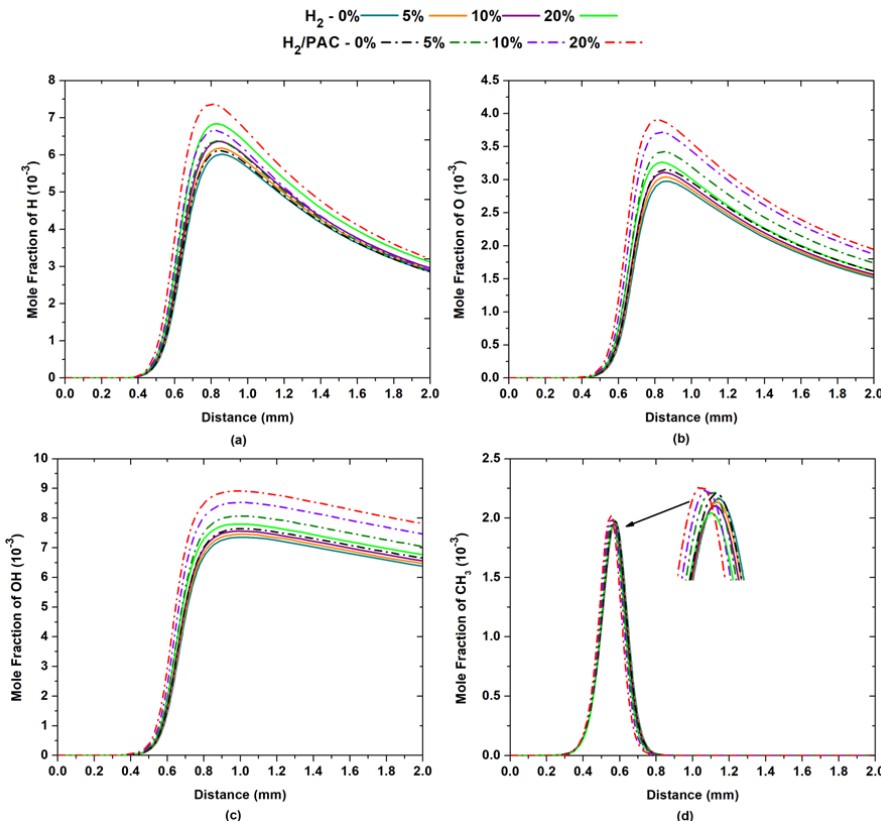

**Figure 10.** Mole fraction profiles of (**a**) H, (**b**) O, (**c**) OH, and (**d**) CH$_3$ with different blends of hydrogen without or with NSPD.

Figure 11 shows the production rate of Equations (6) and (7) with $x_{H2} = 0$ and $x_{H2} = 0.2$ with/without NSPD. The rate increased and the peak shifted upstream with hydrogen addition, but with $x_{H2} = 0.2$ and plasma, a significant impact was seen.

H$_2$ and O$_2$ mole fractions change with hydrogen blends and NSPD, shown in Figure 12. H$_2$ transforms from intermediate species to initial reactant in methane flames with $x_{H2} \geq 0.2$ and NSPD. H$_2$ starts reacting upstream in $x_{H2} = 0.2$, confirmed by [36]. H$_2$ promotes combustion and moves the reaction region towards upstream due to its higher reactivity than CH$_4$.

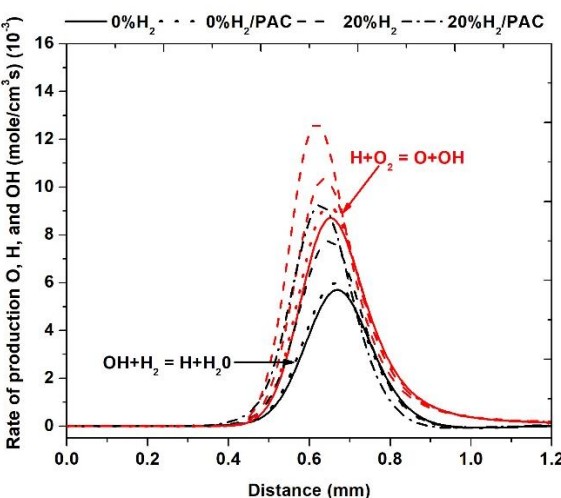

**Figure 11.** Rate of production of O, H, and OH with different blends of hydrogen without or with NSPD.

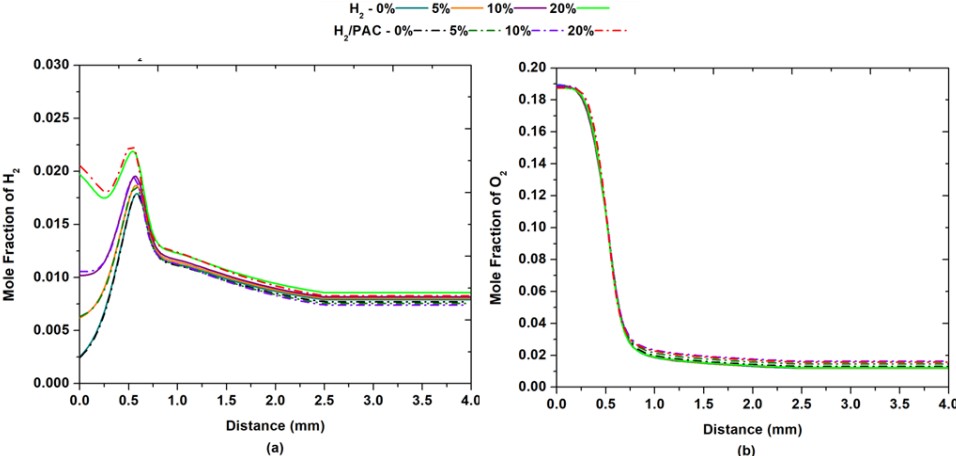

**Figure 12.** Mole fraction profiles of $H_2$ and $O_2$ with different blends of hydrogen without or with NSPD.

Figure 13a illustrates the mole fractions of $CH_4$ in different $H_2$ blends with or without the NSPD. The addition of $H_2$ and NSPD leads to a decrease in $CH_4$ mole fraction, possibly due to the high reactivity of $H_2$ and lower $CH_4$ concentration. The oxidation of $CH_4$ greatly increased and its profiles were shifted towards the upstream sides. $CH_4$ was mainly consumed by reactions with active particles O, H, and OH. The dominant $CH_4$ consumption reactions are listed below.

$$OH + CH_4 = CH_3 + H_2O \tag{8}$$

$$H + CH_4 = CH_3 + H_2 \tag{9}$$

$$O + CH_4 = OH + CH_3 \tag{10}$$

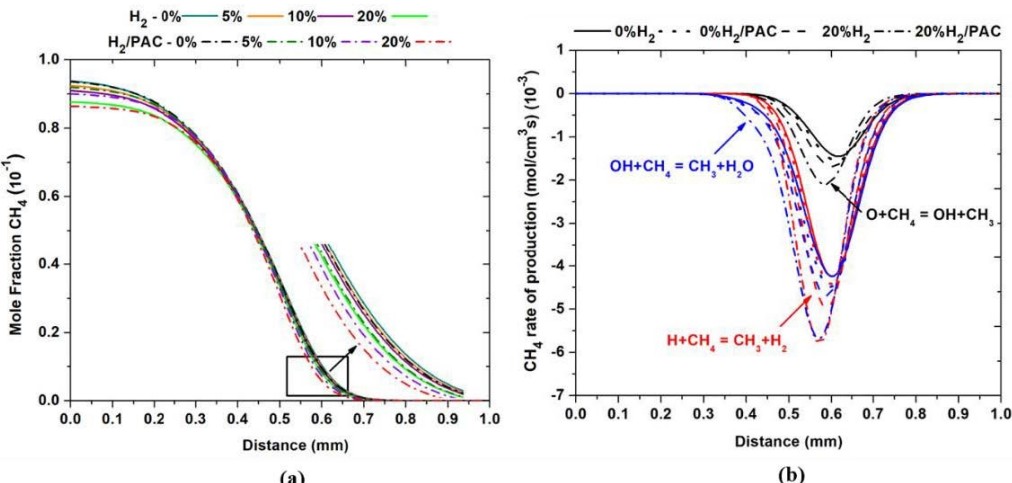

**Figure 13.** (**a**) Mole fraction profile of $CH_4$ (**b**) Rate of production of $CH_4$ with different blends of hydrogen without or with NSPD.

The rate of production of Equations (8)–(10) using hydrogen contents $x_{H2} = 0$ and $x_{H2} = 0.2$ with or without NSPD is shown in Figure 13b. $CH_4$ consumption was increased for reaction Equations (8)–(10) and the peak of the reaction region was shifted towards the upstream with the addition of hydrogen contents with or without plasma. However, when combining the $H_2$ blends of $x_{H2} = 0.2$ with NSPD, a noticeable impact was observed. It was because hydrogen is more reactive, which promoted methane combustion. The concentration of active particles O, H, and OH were increased when methane was blended with hydrogen, mainly due to the chemical effects. Moreover, the NSPD further improved the combustion process due to the thermal (moderate gas heating) and kinetic effects (excitation, ionization and decomposition of fuel and air molecules occurred, which resulted in the production of intermediate fuel fragments and active particles).

Finally, the lean flammability limit is discussed as the minimum equivalence ratio for flame propagation. Figure 14 shows the lean flammability limit using hydrogen contents $X_{H2} = 0$ and $X_{H2} = 0.2$ with or without NSPD. The flammability limit remained at $\phi = 0.6$ without $H_2$ and plasma but improved to $\phi = 0.5$ with the addition of $X_{H2} = 0.2$. Plasma discharge had a significant impact on the flammability limits, with $\phi = 0.45$ at flame temperature about 1500 K.

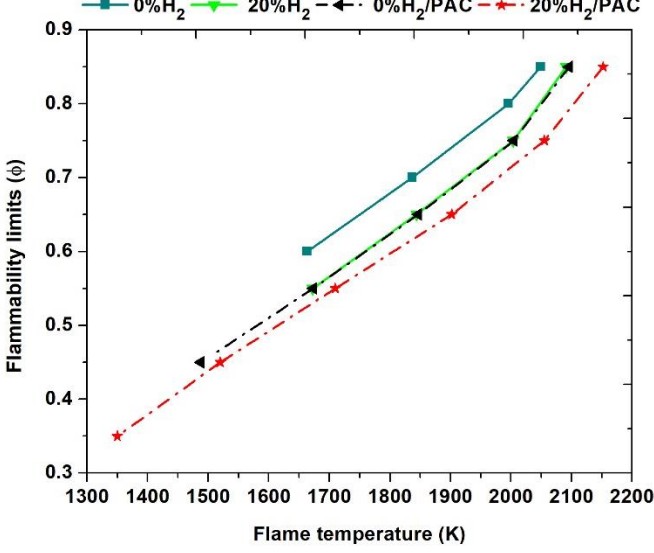

**Figure 14.** Flammability limits with different blends of hydrogen without or with NSPD.

Combining $X_{H2}$ = 0.2 and NSPD increased the flammability limit to $\phi$ = 0.35 at 1350 K, allowing self-sustained combustion at lower flame temperatures and reduced $NO_x$ emissions. The improved flammability limits reduce fuel consumption due to the enhanced reactivity and chemical effects of $H_2$ and thermal and kinetic effects of NSPD.

## 5. Conclusions

This paper investigated the impact of NSPD on enhancing the flame propagation of $CH_4/H_2$/air mixture under ambient temperature and pressure. A reduced electric field experimentally estimated was used for numerical investigation. An extended version of the plasma and combustion kinetic mechanism was applied and validated using available experimental and numerical data. ZDPlasKin was used to predict the temporal evolution of active particles and the results were integrated into CHEMKIN to enhance the flame speed. The numerical study was carried out with varying $H_2$ contents from 0 to 20% in methane/air with or without plasma actuation. It was noticed that with the enrichment of $H_2$ concentration in the methane/air mixture at fixed plasma, the mole fraction of active species was significantly improved. However, the improvements in the production of active particles were linearly increased with the increase in $H_2$ contents. The highest improvement in flame propagation was observed at 20% $H_2$/Plasma reaching 35%.

Flame speed improvement was significantly higher at lean conditions and low flame temperatures. For instance, at an equivalence ratio of 0.8, 20% $H_2$/Plasma resulted in a flame speed of 37 cm/s at a flame temperature of around 2040 K. This same flame speed was also observed in the case of 0% and 5% $H_2$ with a flame temperature close to 2200 K, meaning that high flame speed can be achieved at lean conditions and low flame temperatures, reducing $NO_x$ emissions. Figure 7 shows the comparison of flame speed improvement at lean, stoichiometric, and rich conditions with $x_{H2}$ = 0.2 with or without NSPD. The improvement in flame speed was higher at lean conditions (equivalence ratio of 0.6) with the addition of $x_{H2}$ = 0.2, reaching 15%. Combining $x_{H2}$ = 0.2 with plasma discharge significantly increased flame speed by more than 50%. Furthermore, the combination of $H_2$ blend ($x_{H2}$ = 0.2) and NSPD improved the flammability limit to equivalence ratio 0.35 at a flame temperature of 1350 K, allowing for reduced fuel consumption.

**Author Contributions:** Conceptualization, M.G.D.G.; Methodology, M.G.D.G. and G.M.; Software, G.M.; Validation, G.M.; Formal analysis, G.M.; Investigation, G.M. and S.B.; Data curation, G.M., G.C., Z.A.S. and S.B.; Writing—original draft preparation, G.M.; Writing—review and editing, M.G.D.G.; Supervision, M.G.D.G. and A.F.; Project administration, M.G.D.G. and A.F.; Funding acquisition, G.M. All authors have read and agreed to the published version of the manuscript.

**Funding:** The work was supported and funded by the PON R&I 2014–2020 Asse I "Investimenti in Capitale Umano" Azione I.1 "Dottorati Innovativi con caratterizzazione industriale"—Corso di Dottorato in "Ingegneria dei Sistemi Complessi" XXXV ciclo—Università degli Studi del Salento"— Borsa Codice: DOT1312193 no. 3. This project is also received funding from the Clean Sky 2 Joint Undertaking (JU) under the grant agreement no. 831881 (CHAiRLIFT). The JU received support from the European Union's Horizon 2020 research and innovation program and the Clean Sky 2 JU members other than the Union.

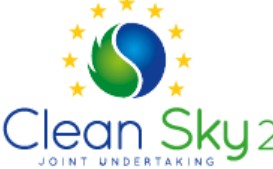
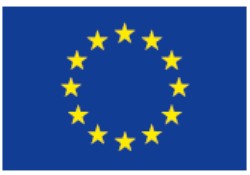

**Conflicts of Interest:** The authors declare no conflict of interest.

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
