# Peer review of "Comparative Analysis of Flame Propagation and Flammability Limits of CH4/H2/Air Mixture with or without Nanosecond Plasma Discharges"

_aerospace, doi:10.3390/aerospace10030224_

Round 1

Reviewer 1 Report

This is an excellent study and worthy for publication.  I have a few comments for the authors’ consideration.  Figures 3, 6, 7 and 9 are valuable and the reduced chemical kinetic model, R1 through R5 seems reasonable.  The authors need to explain why “combining the H2 blends of xH2 = 0.2 with NSPD, a noticeable impact was observed.” which was stated on page 12 last para.  The direct measurement of radicals in flame is not easy to validate the numerical predictions shown in Figures, 5 and 10, but could be important.  It will be a challenge to conduct reliable experiments to control nano second discharge of plasma to flames.  There are experimental reports dedicated to the measurement of C2, CH and other radicals in small methane flames, which may be of interest to the authors in the future.  Z. Diao, et al., Experimental Thermal and Fluid Science, 102: 20-27 (2019). 

Author Response

Reviewer 1 comments

This is an excellent study and worthy of publication.  I have a few comments for the authors’ consideration.  

Figures 3, 6, 7 and 9 are valuable and the reduced chemical kinetic model, R1 through R5 seems reasonable. The authors need to explain why “combining the H2 blends of xH2 = 0.2 with NSPD, a noticeable impact was observed.” which was stated on page 12 last para.

Response: We are thankful for the reviewer comment. It was because hydrogen is more reactive which promoted methane combustion. The concentration of active particles O, H, and OH were increased when methane is blended with hydrogen mainly due to the chemical effects. Moreover, the NSPD further improved the combustion process due to the thermal (moderate gas heating) and kinetic effects (excitation, ionization and decomposition of fuel and air molecules occurred, which resulted in the production of intermediate fuel fragments and active particles).

Above discussion is added in the revised manuscript. Please check the manuscript. Page 12.

The direct measurement of radicals in flame is not easy to validate the numerical predictions shown in Figures, 5 and 10, but could be important. It will be a challenge to conduct reliable experiments to control nano second discharge of plasma to flames. There are experimental reports dedicated to the measurement of C2, CH and other radicals in small methane flames, which may be of interest to the authors in the future.  Z. Diao, et al., Experimental Thermal and Fluid Science, 102: 20-27 (2019). 

Response: The authors thank for the reviewer comment and valuable suggestion. We agreed with the reviewer comments, there are many complexities to conduct the reliable experiments to control the NSPD in flames. Thanks for the suggested study, we will consider it in future for the validation of methane flames. We also validated the plasma kinetic mechanism of methane/air flames with the experimental data available in our previously published study by comparing the decay process of O atoms mole fraction.

“Ghazanfar Mehdi, Donato Fontanarosa, Sara Bonuso and Maria Grazia De Giorgi, Ignition thresholds and flame propagation of methane-air mixture: detailed kinetic study coupled with electrical measurements of the nanosecond repetitively pulsed plasma discharges, 2022 J. Phys. D: Applied Phys. 55 315202”.

Reviewer 2 Report

(1) What the affecting region for plasma to determine flame speed? it needs certain flame range to determine this parameter. The plasma affecting on flame kernal and early stage has different to plasma covering all region.

(2) What the mechanism for flame speed enhancement for plasma? Does it depend on fuel type? H2 addition increases H, O and OH radicals concentration to increase flame speed.

(3) Flammability reflects both ignitability and flame development sustainability. It is necessary to analyze the flame kernal formation and development by plasma introduction.

(4) It will more persusiveness if comparison to other experimental results in the published literatures. This gives strong support to the simulation from quantitative aspect rather than quanlitative aspect without experimental validation by simulation.

Author Response

Reviewer 2 comments

  • What the affecting region for plasma to determine flame speed? it needs certain flame range to determine this parameter. The plasma affecting on flame kernal and early stage has different to plasma covering all region.

Response: The authors thank for the reviewer comment.

We agree that plasma can affect the flame speed in different ways depending on its location and extent in the combustion region. In the early stages of combustion, plasma can affect the ignition and formation of the flame kernel, which is the small, localized region where the flame initiates. The presence of plasma can increase the energy and ionization levels in the kernel, promoting faster reactions and more efficient flame growth. As the flame propagates through the combustion region, plasma can continue to affect the reaction rates and species concentrations, which can in turn influence the overall flame speed.

In this study, we combined a zero-dimensional ZDPlaskin model with a one-dimensional premixed laminar flame speed reactor. For our numerical analysis, we assumed that the non-equilibrium plasma created from a CH4/H2/air mixture at atmospheric pressure is uniformly distributed, which is a similar assumption to what was previously done in [1]. Although the nanosecond pulsed plasma combustion process is three-dimensional and not homogeneous, we used a simplified homogeneous model. To investigate the effects of plasma CH4/H2/air products on flame speed and flammability limits, we used the plasma products of CH4/H2/air as the inlet domain of the reactor. However, we have established that the calculation domain of the CHEMKIN reactor ranges from -2.0 cm upstream to 4.0 cm downstream with respect to the reactor and is sufficient to attain adiabatic equilibrium. This same computation region was also previously utilized in [2].

[1] Y. Wang, P. Guo, H. Chen, Z. Chen, Numerical modeling of ignition enhancement using repetitive nanosecond discharge in a hydrogen/air mixture I: calculations assuming homogeneous ignition, J. Phys. D: Appl. Phys, 54 (2021) 065501 (12pp).

[2]  Yaoyao, Y.; Dong, Liu. Detailed influences of chemical effects of hydrogen as fuel additive on methane flame. International Journal of Hydrogen Energy, 2015, 40(9), 3777-3788.

Above discussions have been added in the revised manuscript.

  • What the mechanism for flame speed enhancement for plasma? Does it depend on fuel type? H2 addition increases H, O and OH radicals concentration to increase flame speed.

Response: We are thankful for the reviewer comment.

The mechanism for flame speed enhancement due to plasma can vary depending on the specific conditions of the combustion system and the plasma discharge. However, some common mechanisms that have been proposed or observed include:

  • Increased ionization and dissociation of the reactant species: Plasma can increase the level of ionization and dissociation of the fuel and oxidizer, which can promote faster reaction rates and more efficient energy release. This can lead to higher flame temperatures, more reactive intermediates, and faster flame propagation.
  • Enhanced radical production and transport: Plasma can generate a variety of reactive species such as radicals, ions, and excited molecules that can participate in combustion chemistry. These species can be transported to the flame front and contribute to the reaction rates and energy release. For example, H2 addition can increase the concentration of H, O and OH radicals, which can promote faster flame propagation and higher flame temperatures.

Particularly, the concentration of active particles O, H, and OH were significantly increased by combining the effects of hydrogen and NSPD. Moreover, the O3 was also increased with the use of NSPD in the mixture of CH4/H2/air (Figure 5 in the paper). It is already stated in the literature that O3 improved the flame speed [3].

[3] Halter, F.; Higelin, P.; Dagaut, P. Experimental and detailed kinetic modelling study of the effect of ozone on the combustion of methane, Energy Fuels, 2011, 25 2909–1.

The extent and importance of these mechanisms may depend on the specific fuel type and combustion conditions. For example, the effect of plasma on flame speed and combustion chemistry may differ for hydrocarbon fuels compared to hydrogen or methane. The plasma parameters, such as the discharge power, frequency, and duration, may also influence the extent and mechanism of flame speed enhancement.

  • Flammability reflects both ignitability and flame development sustainability. It is necessary to analyze the flame kernal formation and development by plasma introduction.

Response: The authors thank for the reviewer comment. We agree that plasma can affect the flame speed in different ways depending on its location and extent in the combustion region. In the early stages of combustion, plasma can affect the ignition and formation of the flame kernel, which is the small, localized region where the flame initiates. The presence of plasma can increase the energy and ionization levels in the kernel, promoting faster reactions and more efficient flame growth. As the flame propagates through the combustion region, plasma can continue to affect the reaction rates and species concentrations, which can in turn influence the overall flame speed.

However, in this work, we coupled 0-D ZDPlaskin with 1-D premixed laminar flame speed reactor. In the present numerical study, it was assumed that the non-equilibrium plasma produced at atmospheric pressure in CH4/H2/air mixture is uniformly distributed (similar assumptions as previously taken by [1]). So even if the nanosecond pulsed plasma combustion process is three-dimensional, rather than homogenous, the simplified homogeneous model is considered. To analyze the impacts of plasma CH4/H2/air products on flame speed and flammability limits, we considered plasma products of CH4/H2/air as an inlet domain of reactor.

The peak concentration of species and reaction rate increased and shifted to the upstream with the addition of hydrogen and NSPD (Figure 10, 11, and 13). It means flame kernal was shifted to the reactor exit.

  • It will more persusiveness if comparison to other experimental results in the published literatures. This gives strong support to the simulation from quantitative aspect rather than qualitative aspect without experimental validation by simulation.

Response: The authors thank for the reviewer comment. We already presented the experimental validation result of plasma kinetic mechanism in our article [4]. In this paper, we presented the experimental validation of chemical kinetic mechanism. However, at present, there is few preliminary experimental study available in the literature related to the nanosecond plasma discharge of CH4/H2/air mixture.

[4] Ghazanfar Mehdi, Donato Fontanarosa, Sara Bonuso and Maria Grazia De Giorgi, Ignition thresholds and flame propagation of methane-air mixture: detailed kinetic study coupled with electrical measurements of the nanosecond repetitively pulsed plasma discharges, 2022 J. Phys. D: Applied Phys. 55 315202.

Reviewer 3 Report

The present paper numerically and theoretically assesses effects of nanosecond pulse discharge on flame propagation for a CH4/H2/air mixture, relevant to design and operation of low-emission combustion systems. Plasma kinetics using ZDPlaskin and BOLSIG+ is coupled with the detailed combustion kinetic mechanisms for combustion of CH4 and H2 to compute the laminar flame speed. This is a complicated effort as the plasma-assisted combustion kinetics for oxidation of hydrocarbons is seldom developed. 

It seems to this reviewer that effects of NSPD are incorporated into the flame-propagation model by setting the value for the reduced electric field. In real applications, plasma kernels are only initiated at isolated narrow regions. However, this reviewer understands that this is a 1D simulation for the laminar flame speed. Simplifications and idealizations are warranted.   

Author Response

Reviewer 3 comments

The present paper numerically and theoretically assesses effects of nanosecond pulse discharge on flame propagation for a CH4/H2/air mixture, relevant to design and operation of low-emission combustion systems. Plasma kinetics using ZDPlaskin and BOLSIG+ is coupled with the detailed combustion kinetic mechanisms for combustion of CH4 and H2 to compute the laminar flame speed. This is a complicated effort as the plasma-assisted combustion kinetics for oxidation of hydrocarbons is seldom developed. 

It seems to this reviewer that effects of NSPD are incorporated into the flame-propagation model by setting the value for the reduced electric field. In real applications, plasma kernels are only initiated at isolated narrow regions. However, this reviewer understands that this is a 1D simulation for the laminar flame speed. Simplifications and idealizations are warranted.   

Response: The authors thanks for the reviewer comments and recommendation for publication.

Round 2

Reviewer 2 Report

The revised version can be accepted for publication.